# Anti-PD1 and Anti-PDL1-Induced Hypophysitis: A Cohort Study of 17 Patients with Longitudinal Follow-Up

**DOI:** 10.3390/jcm9103280

**Published:** 2020-10-13

**Authors:** Manon Levy, Juliette Abeillon, Stéphane Dalle, Souad Assaad, Françoise Borson-Chazot, Emmanuel Disse, Gérald Raverot, Christine Cugnet-Anceau

**Affiliations:** 1Faculté de Médecine, Université Lyon 1, 69008 Lyon, France; manon.levy@chu-lyon.fr (M.L.); stephane.dalle@chu-lyon.fr (S.D.); francoise.borson-chazot@chu-lyon.fr (F.B.-C.); emmanuel.disse@chu-lyon.fr (E.D.); gerald.raverot@chu-lyon.fr (G.R.); 2Fédération d’Endocrinologie, Centre de Référence Maladies Rares hypophysaires, Groupement Hospitalier Est, Hospices Civils de Lyon, 69500 Bron, France; juliette.abeillon@chu-lyon.fr; 3Service d’Endocrinologie, Diabète, Nutrition, Centre Hospitalier Lyon Sud, Hospices Civils de Lyon, 69310 Pierre-Bénite, France; 4ImmuCare, Institut de Cancérologie, Hospices Civils de Lyon, 69002 Lyon, France; 5Tox’imm, Centre Léon Bérard, 69008 Lyon, France; souad.assaad@lyon.unicancer.fr; 6Service de Dermatologie, Centre Hospitalier Lyon Sud, Hospices Civils de Lyon, 69310 Pierre-Bénite, France; 7Service d’Hématologie et Médecine Interne, Centre Léon Berard, 69008 Lyon, France; 8INSERM U1060, INRA 1397, INSA Lyon, Centre de Recherche en Nutrition Humaine Rhône-Alpes (CRNH RA), CarMeN Laboratory, 69310 Pierre-Bénite, France; 9INSERM U1052, CNRS, UMR5286, Cancer Research Center of Lyon, 69008 Lyon, France

**Keywords:** hypophysitis, adrenal insufficiency, immunotherapy, nivolumab, pembrolizumab, programmed cell death 1 protein, programmed cell death 1 ligand 1 protein

## Abstract

Hypophysitis, secondary to programmed cell death 1 protein (PD1) and programmed cell death 1 ligand 1 (PDL1) inhibitors, were thought to be rare, with only a few studies describing more than one case with long-term follow-up. The aim of the present study was to describe the clinical, laboratory, and morphological characteristics of PD1/PDL1 inhibitor-induced hypophysitis, and its long-term course. This cohort study was conducted at the University Hospital of Lyon, France, with longitudinal follow-up of patients. Seventeen cases of PD1/PDL1 inhibitor-induced hypophysitis were included. The median time to onset of hypophysitis was 28 weeks (range: 10–46). At diagnosis, 16 patients complained of fatigue, 12 of nausea or loss of appetite, while headache was rare. We found no imaging pituitary abnormality. All patients presented adrenocorticotropic hormone (ACTH) deficiency; other pituitary deficiencies were less common (*n* = 7). At last follow-up (median: 13 months), ACTH deficiency persisted in all but one patient and one patient recovered from gonadotropic deficiency. PD1/PDL1 inhibitor-induced hypophysitis is a clinical entity different from those associated to cytotoxic T-lymphocyte antigen-4 (CTLA4) inhibitors, with less obvious clinical and radiological signs, and probably a different mechanism. The paucity of symptoms demonstrates the need for systematic hormonal follow-up for patients receiving PD1/PDL1 inhibitors.

## 1. Introduction

Immune checkpoint inhibitors (ICI) target key points in the immune system’s tolerance to tumors in order to enhance the anti-tumor response, such as cytotoxic T lymphocyte antigen 4 (CTLA4), programmed cell death 1 protein (PD1) and its ligand programmed cell death 1 ligand 1 (PDL1) [1,2,3]. Because ICI modulate the immune system, they are responsible for immune-related adverse events (IRAEs), including frequent dysfunction of the endocrine system. The most frequent endocrine adverse event is thyroid dysfunction; it is estimated that 6.5% of patients treated with ICI developed hypothyroidism and 2.9% developed hyperthyroidism, and this is more common among patients treated with PD1 or PDL1 inhibitors [4]. The second most frequent is hypophysitis, which has been reported in between 5.6% and 13.6% of patients treated with the CTLA4 inhibitor ipilimumab [5,6], but is less common with other ICI: the reported frequency is between 0.5 and 1.1% for PD1 inhibitors [5,6], and less than 0.1% for PDL1 inhibitors [4]. Combination therapy, mostly the association of PD1 and CTLA4 inhibitors, has been associated with the most frequent occurrence of hypophysitis, ranging from 8.8 to 10.5% [5]. However, estimations of the frequency of hypophysitis occurrence are probably under-estimated in oncological trials that did not systematically screen for this adverse event that has non-specific clinical presentation, and this is supported by results of studies that included a specific hormonal screening [6].

CTLA4 inhibitor-induced hypophysitis is now well described. The descriptions of PD1 and PDL1 inhibitor-induced hypophysitis are more recent, probably because these drugs have been approved more recently than the approval of CTLA4 inhibitor. Some case reports described isolated adrenocorticotropic hormone (ACTH) deficiency associated with PD1 and PDL1 inhibitors [7,8,9,10,11,12,13,14,15,16,17,18].

Based on our clinical experience and observations reported in the literature, we hypothesized that PD1/PDL1 inhibitor-induced hypophysitis would have different characteristics compared to CTLA4 inhibitor-induced hypophysitis. There are few published data on the hormonal follow-up of such patients and recovery of pituitary deficiencies. The aim of the present study was therefore to describe the clinical, laboratory, and morphological characteristics of PD1/PDL1 inhibitor-induced hypophysitis as well as its long-term clinical course.

## 2. Materials and Methods

We conducted a descriptive cohort study with longitudinal follow-up, in two referral centers of the University Hospital of Lyon (Hospices Civils de Lyon), France. Adult patients referred by oncologists to endocrinologists for suspicion of hypophysitis due to PD1 or PDL1 inhibitors from 29 December 2015 to 21 June 2019 were included. Patients were referred to endocrinologists through the ImmuCare and Tox’imm networks (healthcare networks in Lyon including patients who have presented IRAEs due to ICI in the treatment of cancers). In addition, the ImmuCare database was also screened for other eligible patients, using the terms “hypophysitis” and “adrenal insufficiency”. Patients receiving a combination of PD1 and CTLA4 inhibitors, or with a history of previous intake of CTLA4 inhibitor were excluded, as were patients who had received corticosteroids in the previous six months or with a history of cranial radiotherapy. Medical records, including clinical examination, medication, laboratory and radiological results, were reviewed in detail. Data were processed anonymously. In accordance with legislation in place at the time of the study, it was declared to the data protection agency (Commission Nationale Informatique et Liberté; no. 18-297) and an information letter was sent to patients—informed consent was not required. The study was also approved by the Ethics Committee of the Hospices Civils de Lyon (no. 19–163).

The diagnosis of ICI-induced hypophysitis was presumptive, based on the presence of a new pituitary deficiency, associated or not with clinical symptoms and radiographic pituitary enlargement, in the absence of an alternative etiology such as previous corticosteroid intake, history of cranial radiotherapy, history of systemic inflammatory disease, and pituitary metastasis on magnetic resonance imaging (MRI) [6,19]. No biopsy was available as no patient underwent pituitary surgery or autopsy. The severity grade was defined according to the Common Terminology Criteria for Adverse Events (CTCAE). Pituitary MRI results were included if performed within two months following the clinical diagnosis.

All patients had a hormonal evaluation at diagnosis, including 8 am cortisol and ACTH, thyroid-stimulating hormone (TSH), free thyroxin (T4), free 3,5,3′-triiodothyronine (T3), follicle-stimulating hormone (FSH) and luteinizing hormone (LH), total testosterone for males, prolactin, and insulin-like growth factor 1 (IGF1). Serum hormone measurements were performed as follows: cortisol measurement was performed by chemiluminescence immune assay (I2000; Abbott laboratories, Abbott Park, IL, USA and Aia 360; Tosoh Bioscience, Tokyo, Japan) or electrogenerated chemiluminescence (Elecsys; Roche Diagnostics, Mannheim, Germany); ACTH measurement was performed by chemiluminescence immune assay (Liaison XL; DiaSorin, Saluggia, Italy) or electrogenerated chemiluminescence (Cobas e601; Roche Diagnostics, Mannheim, Germany); TSH, free T4, free T3, FSH and LH measurements were performed by chemiluminescence immune assay (I2000; Abbott laboratories, Abbott Park, IL, USA); total testosterone measurement was performed by an in-house radioimmunoassay with extraction and chromatography; prolactin measurement was performed by electrogenerated chemiluminescence (Cobas e601; Roche Diagnostics, Mannheim, Germany); IGF1 measurement was performed by chemiluminescence immune assay (ISYS; Immunodiagnostics Systems, Boldon, UK. ACTH deficiency was defined as a cortisol level at 8 am below 138 nmol/L, with ACTH below or within the reference range (10–50 pg/mL), according to current French guidelines [20]. Thyrotropic deficiency was defined as free T4 below the lower limit of the laboratory reference range, with normal or low TSH value. In males, gonadotropic deficiency was defined as a total testosterone below the laboratory reference range, with normal or low FSH and LH. In menopausal women, gonadotropic deficiency was defined as normal or low FSH and LH. Somatotropic deficiency was suspected when IGF1 was below the laboratory reference range for age, but no growth hormone (GH) stimulation testing was performed to confirm diagnosis because GH replacement would be contraindicated in patients with an active malignancy.

Frequency of hormonal follow-up and attempts to wean hormonal replacement following the diagnosis of hypophysitis were at the physician’s discretion. Follow-up included clinical examination and hormone tests including 8 am cortisol, 8 am ACTH if available, TSH, free T4, FSH and LH, total testosterone for males, and IGF1 if available. Recovery of ACTH function was suspected if 8 am cortisol level was greater than 138 nmol/L without intake of oral hydrocortisone or corticosteroids, and confirmed by an ACTH stimulation test finding an increase in cortisol value greater than the laboratory threshold one hour after an infusion of 0.25 mg of tetracosactide (synacthen test). Thyrotropic recovery was defined as free T4 and TSH within the laboratory reference range without hormonal substitution. In males, gonadotropic recovery was defined as total testosterone, FSH, and LH within the laboratory reference range without testosterone intake. In menopausal women, gonadotropic recovery was defined as FSH and LH greater than the laboratory reference range.

## 3. Results

### 3.1. Patients

A total of 25 patients were referred to endocrinologists or were included via the ImmuCare database. One patient who had primary adrenal insufficiency and seven who had received corticosteroids in the previous six months were excluded; a total of 17 cases of PD1/PDL1 inhibitor-induced hypophysitis were analyzed (Figure 1).

The mean ± standard deviation age was 64 ± 8.2 years, the majority of patients were men (76.5%), and melanoma was the main type of cancer (47.1%). Sixteen patients received PD1 inhibitors (94.1%) and one received PDL1 inhibitor (5.9%; Table 1).

### 3.2. Presentation at Diagnosis

The median time to onset of hypophysitis was 28 weeks (range: 10–46) after ICI initiation and this occurred after a median of nine courses (range: 3–21). The signs that led to the diagnosis of hypophysitis were clinical symptoms (fatigue, nausea and/or loss of appetite) in most cases (*n* = 13, 76.5%), hyponatremia leading to cortisol measurement in one case (5.9%), and hormonal abnormalities found on systematic monitoring in three cases (17.6%). At diagnosis, 16 patients complained of fatigue, 12 patients complained of nausea and/or loss of appetite, three patients complained of headache, and one patient was asymptomatic. Hyponatremia was present in nine cases at diagnosis (52.9%). A pituitary MRI within two months after diagnosis was available for 14 patients; no MRI showed pituitary enlargement or other pituitary abnormality (Table 2).

All patients presented ACTH deficiency (Table 2). The median value of cortisol at diagnosis was 25 nmol/L (interquartile range, IQR: (0–42)). Two patients presented thyrotropic deficiency (11.8%). Three patients presented gonadotropic deficiency (18.8%), all were male and did not have simultaneous elevation of prolactin. IGF1 was low for two patients (13.3%; Table 2). Prolactin at diagnosis was available for 15/17 patients; it was within the laboratory reference range in seven cases, and moderately elevated in eight cases (47.1%; range: 26.6–51.3 µg/L). No patient presented diabetes insipidus at diagnosis.

Most cases (94.1%) of hypophysitis were of severity grade 2 or 3. Three patients died during follow-up from causes unrelated to hypophysitis; one died four months after diagnosis due to thrombotic microangiopathy, and the two others died 10 months after diagnosis, one due to peritonitis related to abdominal metastases and one due to gastrointestinal bleeding. For four patients a planned course of ICI was delayed, but hypophysitis was not responsible for permanent discontinuation of ICI (Table 2).

### 3.3. Treatment

All patients received treatment for their ACTH deficiency. Seven patients with grade 3 hypophysitis were hospitalized and received initial intravenous high-dose hydrocortisone (single injection of 100 mg of intravenous hydrocortisone, followed by oral hydrocortisone in a progressively decreasing dose). One patient received high-dose oral prednisolone for severe headache. After the acute period following diagnosis, all patients were treated with oral hydrocortisone at replacement doses (15 to 20 mg per day). The two patients with thyrotropic deficiency were treated with levothyroxin. Of the three patients with gonadotropic deficiency, no patient was treated with testosterone. When somatotropic deficiency was suspected by low IGF1, no treatment was introduced.

### 3.4. Clinical and Laboratory Course

The median follow-up period was 13 months (IQR: (10.75–29.250)). Clinical and laboratory follow-up was available for 16 patients; one patient died four months after diagnosis without hormonal follow-up.

One patient recovered his ACTH function two months after diagnosis. This patient was a 52-year-old man treated for metastatic melanoma with pembrolizumab every three weeks at a dose of 2 mg/kg, a regimen similar to other patients treated for melanoma. He presented a grade 1 hypophysitis after four cycles of pembrolizumab. The diagnosis was made on a systematic monitoring that found an 8 am cortisol of 30 nmol/L and an 8 am ACTH of 13.2 pg/mL. The patient was asymptomatic. He had no other associated pituitary deficiency and presented no other IRAE. No pituitary MRI was available for this patient. Two months after diagnosis, hormone tests found an 8 am cortisol of 309 nmol/L before hydrocortisone intake. The patient responded to an ACTH stimulation test and oral hydrocortisone therapy was stopped. Oral hydrocortisone was continued at the end of follow-up for all other patients. Regarding the other axes, among the three patients with gonadotropic deficiency, one patient recovered, one had a persistent deficiency, and for the third no testosterone was available during follow-up. Among the two patients with thyrotropic deficiency, both had a persistent deficiency. No follow-up was available for the two patients with low IGF1 regarding this hormone.

### 3.5. Other Endocrine IRAEs

Four patients presented associated primary thyroid dysfunction induced by ICI; three patients presented new-onset hypothyroidism and one presented thyrotoxicosis progressing to hypothyroidism.

In two cases thyroid dysfunction occurred first, one and six months before the diagnosis of hypophysitis. In one case thyroid dysfunction occurred four months after diagnostic of hypophysitis. In one case hypophysitis and thyroid dysfunction were diagnosed at the same time.

All patients received levothyroxin replacement therapy that was not discontinued at the end of the follow-up.

## 4. Discussion

In the present study, PD1/PDL1 inhibitor-induced hypophysitis was associated with ACTH deficiency at diagnosis in all cases and multiple deficiencies were less common. In most cases, the clinical presentation was not striking, as most patients complained of fatigue, nausea and/or loss of appetite; conversely, headache was rare and no patient complained of vision disorder. Additionally, MRI was not contributive to a positive diagnosis as none showed pituitary abnormality, although it remains necessary to rule out differential diagnoses and in particular pituitary metastasis.

The clinical characteristics are similar to those previously described regarding PD1/PDL1 inhibitor-induced hypophysitis, in particular those recently reported by Faje et al. in a cohort of 22 PD1 inhibitor-induced hypophysitis [6]. In contrast to that found herein, a pituitary enlargement on MRI was reported in 28% of cases by Faje et al. [6], considering MRIs performed within a month after diagnosis. As a previous study found that pituitary gland enlargement resolved within a month in approximately half of patients with ipilimumab-associated hypophysitis [21], and we considered MRIs performed up to two months, the latest MRIs could have missed the enlargement phase (three MRIs in the present study were performed after one month). However, it can be concluded that MRI abnormality is less common in PD1/PDL1 inhibitor-induced hypophysitis than in CTLA4 inhibitor-induced hypophysitis.

Given these results, PD1/PDL1 inhibitor-induced hypophysitis seems to have different characteristics and mechanism compared to those induced by CTLA4 inhibitors. The latter are now well described, especially those due to ipilimumab which is widely used to treat advanced melanoma. In such cases, headache is quite frequent, reported for about 80–85% of patients [22,23], and the frequency of pituitary enlargement is reported to range from 60 to 100% [23]. Moreover, the distribution of pituitary deficiencies is different. There are usually multiple pituitary deficiencies and frequent, but not systematic, ACTH deficiency [22,24]. Hypophysitis also seems to occur earlier with the use of CTLA4 inhibitors, between six and 11 weeks after treatment initiation [22], while the diagnosis was made between 10 and 46 weeks herein, and between 18 and 44 weeks in the study reported by Faje et al. [6].

The pathophysiology of CTLA4 inhibitor-induced hypophysitis is linked to the expression of the CTLA4 antigen on pituitary endocrine cells (predominantly on prolactin and TSH-secreting cells) and the development of anti-pituitary antibodies (especially against TSH-secreting cells, and to a lesser extent FSH and ACTH-secreting cells), leading to the activation of the classical complement pathway; anti-pituitary antibodies are not present prior to CTLA4 inhibitor treatment, suggesting that the CTLA4 inhibitor can induce both cellular and humoral immune responses against the anterior pituitary [25,26]. The reasons why PD1 and PDL1 inhibitors can also be responsible for this pituitary side effect remains unclear, and the pathophysiology is probably different. Antibodies directed against PD1 and PDL1 are immunoglobulin G4 (IgG4), which, unlike immunoglobulin G1 (IgG1) CTLA4 inhibitors, cannot activate the classical complement pathway [24]. One hypothesis is that ACTH deficiency in PD1/PDL1-induced hypophysitis could be related to the development of antibodies directed against ACTH, triggered by the introduction of a treatment that enhances the immune system, in patients presenting a tumor that expresses ACTH. This mechanism would be similar to that described by Bando et al. who found an ectopic and specific ACTH expression in tumor cells, triggering autoimmunity to corticotroph cells that caused ACTH deficiency, described as a paraneoplastic syndrome in neuroendocrine lung carcinoma [27]. It is also further supported by Fujita et al. who found anti-corticotroph antibodies in 58% of patients presenting acquired isolated ACTH deficiency [28]. Another interesting point is that PD1 inhibitor-induced hypophysitis could be linked to a human leucocyte antigen (HLA) predisposition. Indeed, Inaba et al. described certain HLA types that are more common in this population compared to controls and compared to patients presenting idiopathic isolated corticotroph deficiency [29].

The paucity and non-specificity of symptoms seems to be responsible for delayed diagnosis of PD1/PDL1 inhibitor-induced hypophysitis, or sometimes misdiagnosis both in previous reports [9,12,14,15,17] and herein; this point is illustrated in the present study by fortuitous diagnosis via systematic laboratory monitoring for three of the patients. The correct diagnosis is, however, crucial since pituitary deficiencies can affect quality of life, tolerance to cancer treatment, or even be life-threatening in the case of ACTH deficiency in already fragile patients. A systematic and regular laboratory hormonal follow-up including 8 am cortisol assessment for patients receiving ICI is therefore necessary, not only for patients treated with CTLA4 inhibitors but also for those treated with PD1/PDL1 inhibitors, even if hypophysitis remains less frequent in such patients. The French society of endocrinology published guidelines in 2018 for the endocrinological follow-up of patients receiving ICI, recommending a systematic laboratory evaluation including fasting venous glycemia, natremia, TSH, free T4, 8 am cortisol and testosterone in males, at each course of ICI during the first six months of treatment, and then every two courses during the following six months [30]. After this one-year period, laboratory evaluation is recommended in case of clinical signs [30], but we believe that clinicians should remain vigilant after this period because such signs may be difficult to detect. Furthermore, there is no recommendation regarding follow-up after discontinuation of ICI although some cases have occurred after the discontinuation [10,12,15].

The present study has, however, several limitations. Only one patient of the cohort received PDL1 inhibitor. This is related to the fact that the majority of patients were treated for melanoma, for which only PD1 inhibitors were approved in France at the time of the study. This case shows that hypophysitis can also occur with PDL1 inhibitors, and the clinical presentation is similar to PD1 inhibitor-induced hypophysitis, both in previous reports [18] and herein. Another point is that follow-up data were not exhaustive for all patients, and given the small number of patients with deficiencies other than ACTH deficiency, it is difficult to conclude on the prognosis for recovery of these axes. However, there was no missing data for ACTH among patients with follow-up, and ACTH deficiency was mostly definitive. This is not always the case as there are several reported cases of ACTH recovery: one of the cases herein recovered normal ACTH function two months after diagnosis, Faje et al. described two cases of recovery of ACTH deficiency induced by ipilimumab and one by combination therapy [6,22], and Ryder et al. described three cases of recovery of ACTH deficiency induced by ipilimumab [31]. It is of note that the case reported herein is, to our knowledge, the first related to a PD1 inhibitor, suggesting that this is possible at least in some patients, and could be underestimated since monitoring of the ACTH axis is not carried out routinely after the diagnosis of hypophysitis. We therefore suggest a regular follow-up of 8 am cortisol, before hydrocortisone intake, to diagnose ACTH recovery, every three months during the first year after diagnosis of hypophysitis, and then every six months during the next one year period.

## 5. Conclusions

PD1/PDL1 inhibitor induced-hypophysitis seems to be a clinical entity different from those associated to CTLA4 inhibitors, with less obvious clinical and radiological signs, and probably a different mechanism. The paucity of clinical presentation can lead to a delayed diagnosis and demonstrates the need for systematic and regular hormonal evaluation for patients treated with ICI, especially cortisol screening.

## Figures and Tables

**Figure 1 jcm-09-03280-f001:**
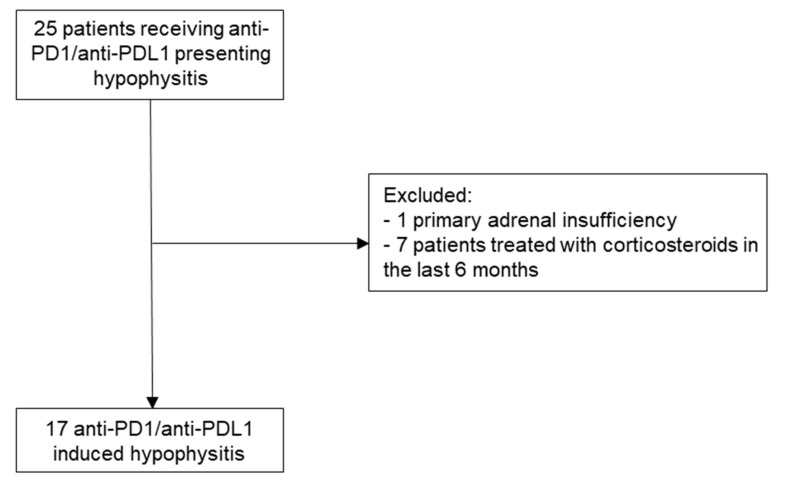
Selection process to include cases. PD1: programmed cell death 1 protein, PDL1: programmed cell death 1 ligand 1.

**Table 1 jcm-09-03280-t001:** Patient characteristics (*n* = 17).

Mean age, years ± standard deviation	64 ± 8.2
Male sex, *n* (%)	13 (76.5)
Type of cancer, *n* (%)	
Melanoma	8 (47.1)
Non-small-cell lung carcinoma	4 (23.5)
Hepatocellular carcinoma	2 (11.8)
Renal carcinoma	2 (11.8)
Cutaneous T-cell lymphoma	1 (5.9)
Type of ICI, *n* (%)	
Anti-PD1	16 (94.1)
Nivolumab	6 (35.3) ^a^
Pembrolizumab	9 (52.9) ^a^
Tislelizumab	1 (5.9) ^a^
Anti-PDL1	1 (5.9)
Atezolizumab	1 (5.9) ^a^
Associated treatment, *n* (%)	
Anti-VEGF antibody	2 (11.8)
Axitinib	1 (5.9)
Epacadostat	1 (5.9)
Brentuximab vedodin	1 (5.9)
History of autoimmune disease, *n* (%)	0 (0)
History of other IRAEs, *n* (%)	5 (29.4)
Thyroid dysfunction	3 (17.6)
Vitiligo	1 (5.9)

ICI: immune checkpoint inhibitors, PD1: programmed cell death 1 protein, PDL1: programmed cell death 1 ligand 1, VEGF: vascular endothelial growth factor, IRAEs: immune related adverse events.^a^ percent among all ICI.

**Table 2 jcm-09-03280-t002:** Description of hypophysitis at diagnosis (*n* = 17).

Symptoms at Diagnosis, *n* (%)	
Fatigue	16 (94.1)
Nausea, loss of appetite	12 (70.6)
Headache	3 (17.6)
Vision disorders	0 (0)
Asymptomatic	1 (5.9)
Hyponatremia, *n* (%)	9 (52.9)
MRI showing abnormalities (among those with available data), *n* (%)	0/14 (0)
Median time to onset after treatment initiation (range; weeks)	28 (10–46)
Median number of courses before diagnosis (range)	9 (3–21)
CTCAE grade, *n* (%)	
1	1 (5.9)
2	9 (52.9)
3	7 (41.2)
4–5	0 (0)
Cycle of ICI delayed due to hypophysitis, *n* (%)	4 (23.5)
ICI discontinued due to hypophysitis, *n* (%)	0 (0)
Pituitary deficiencies at diagnosis (among those with available data), *n* (%)	
ACTH deficiency	17/17 (100)
Thyrotropic deficiency	2/17 (11.8)
Gonadotropic deficiency	3/16 (18.8)
Low IGF1	2/15 (13.3)

MRI: magnetic resonance imaging, CTCAE: common terminology criteria for adverse events, ICI: immune checkpoint inhibitors, ACTH: adrenocorticotropic-hormone, IGF1: insulin-like growth factor 1.

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
