# Peer review of "Anti-PD1 and Anti-PDL1-Induced Hypophysitis: A Cohort Study of 17 Patients with Longitudinal Follow-Up"

_jcm, 2020, doi:10.3390/jcm9103280_

Round 1

Reviewer 1 Report

Manon Levy et al. describes a cohort of 17 cancer patients with hypophysitis induced by anti-PD1 and anti-PDL1-therapy. The research question is whether anti-PD1 and anti-PDL1-induced hypophysitis is different from hypophysitis induced by CTLA4 inhibitors. Briefly, the authors show that PD1/PDL1 inhibitors induced hypophysitis is clinically different from CTLA4 hypophysitis. This is a well written manuscript. I have just one minor point.

Please control page 4, line 142. Hyponatriemia is mentioned in one case and afterwards it is stated that 9 patients had hyponatriemia at the time of diagnosis.

Author Response

Reviewer 1

Manon Levy et al. describes a cohort of 17 cancer patients with hypophysitis induced by anti-PD1 and anti-PDL1-therapy. The research question is whether anti-PD1 and anti-PDL1-induced hypophysitis is different from hypophysitis induced by CTLA4 inhibitors. Briefly, the authors show that PD1/PDL1 inhibitors induced hypophysitis is clinically different from CTLA4 hypophysitis. This is a well written manuscript. I have just one minor point.

Please control page 4, line 142. Hyponatriemia is mentioned in one case and afterwards it is stated that 9 patients had hyponatriemia at the time of diagnosis.

Response to Reviewer 1:

Dear Reviewer,

Thank you for this comment. At the first mention of “hyponatremia” line 167, it was this biological anomaly that led to a control of 8 am cortisol and to the diagnosis of hypophysitis. We have modified the manuscript as follow to make it more understandable (line 167): “The signs that led to the diagnosis of hypophysitis were clinical symptoms in most cases (76.5%), hyponatremia leading to cortisol measurement in one case (5.9%)”.

At line 170 “hyponatremia was present in nine cases at diagnosis”, nine is the total number of patients with hyponatremia, but hyponatremia was the element leading to diagnosis in only one case.

Reviewer 2 Report

  1. There was no comment on the diagnostic methods for hypophysitis itself. And the other work-up to rule out other causes of hypophysitis should be described in details. 
  2. Authors did not present the overall population of patients who took the PD1/PDL1 inhibitors. I suppose the readers are curious about the incidence of PD1/PDL1 inhibitos-induced hypophysitis.
  3. The results of follow-up MR were not described.

Author Response

Reviewer 2

  • There was no comment on the diagnostic methods for hypophysitis itself. And the other work-up to rule out other causes of hypophysitis should be described in details.
  • Authors did not present the overall population of patients who took the PD1/PDL1 inhibitors. I suppose the readers are curious about the incidence of PD1/PDL1 inhibitos-induced hypophysitis.
  • The results of follow-up MR were not described.

Response to Reviewer 2:

Dear Reviewer,

Thank you for your pertinent comments and suggestions. Please find bellow our point-by-point responses.

  • The diagnosis of hypophysitis was based on the presence of a new pituitary deficiency revealed on hormone assays, associated or not with clinical symptoms and pituitary enlargement on MRI. We have used the same definition as the authors of references 6 and 19. Other causes of hypophysitis were ruled out by excluding patients who have received corticosteroids in the previous six months and patients with a history of cranial radiotherapy, as corticosteroids and cranial radiotherapy could have been the cause of the pituitary deficiency. Pituitary MRI ruled out a pituitary metastasis. At least, the patients did not have any systemic inflammatory disease that could be responsible for hypophysitis. However, the diagnosis of ICI-induced hypophysitis remained presumptive.

We have modified the manuscript to detail other causes of hypophysitis ruled out (lines 86 to 90): “The diagnosis of ICI-induced hypophysitis was presumptive, based on the presence of new pituitary deficiency, associated or not with clinical symptoms and radiographic pituitary enlargement, in the absence of an alternative etiology such as previous corticosteroids intake, history of cranial radiotherapy, history of systemic inflammatory disease, pituitary metastasis on magnetic resonance imaging (MRI) [6,19].”

  • Unfortunately, owing to the retrospective design of the study, we were not able to calculate a reliable incidence of PD1/PDL1 inhibitor-induced hypophysitis. We were unable to obtain the exact number of patients receiving PD1/PDL1 inhibitors in our two centers over the entire inclusion period. The ImmuCare database that we used to include some patients who would not have been referred to endocrinologists was created in 2019, which did not allow us to make an exhaustive collection of all the patients treated.

We have calculated an estimated incidence in one of our centers over the year 2018, year in which we were able to collect the most complete list of patients treated. We found a prevalence of 1.1% of PD1 inhibitor-induced hypophysitis (365 patients treated with PD1-inhibitors and 4 cases of hypophysitis).

However, this is only an estimate, and we have considered the calculation of incidence over the entire inclusion period too imprecise to present this result.

  • No pituitary MRI was performed during follow-up. The present study was a descriptive study without follow-up protocol, the examinations performed during follow-up were left to the discretion of the physician in charge of the patient. As the MRI at diagnosis was normal in every patient, physicians did not consider necessary to monitor the pituitary MRI during follow-up.

Reviewer 3 Report

In their manuscript, Levy at al present 17 cases of ICI-induced hypophysitis and suggest means for diagnosis and follow up of ICI-treated cancer patients. The manuscript is interesting and of significant clinical utility. In general it is well structured, although I would definitely suggest that a native English speaker reviews it. Some comments follow.

Major comments

  1. Paragraph 2 in Discussion is quite confusing. I am not sure that the conclusive statement (lines 213-214) is correct, 28% of cases with MRI +/ve results is not negligible. Pls rephrase the entire paragraph and definitely change last sentence.
  2. Do the authors think (and accordingly suggest) that MRIs are useless, else not recommended, in the clinical diagnosis of hypophysitis? Because this is what I understand in paragraph lines 244-260. If so, you need to strongly support this conclusion, with more clinical data and the literature.
  3. Lines 262-264: PDL1 inhibitors are more frequently used than in 1 out of 17 patients. This depends on the type of cancer and the cohort. Pls rephrase. Similarly, I am not convinced that PD1 and PDL1 induced hypophysitis share the same mechanism. These ICIs target completely different cells, signaling Is not the same. Pls correct.
  4. The patient who recovered ACTH deficiency was administered the same dose and scheme of anti-PD1 treatment? Which ICI was used in this particular patient? This may be of relevance and linked to recovery. If possible, compare your data with those in refs 6 and 31.

Minor comments

  1. Line 45, replace the word including
  2. In Materials and Methods add how you determined the hormones listed or include a ref with details on the methodology used.
  3. 1 is not a flow chart, I would say it is the process you followed to select cases reported here
  4. Line 139, pls specify clinical symptoms
  5. Pls look at and correct: line 26, was thought; line 29, 17 cases is not a large cohort, pls change wording; line 36, inhibitor-induced; line 49, treated with; line 60, is now well described; line 62, inhibitor (only 1 approved); line 78, combination; line 89, is it biopsy?; line 91, within two months; line 152, of severity grade; line 170, following diagnosis; line 195, after diagnosis; line 200, in all cases instead of always; line 228, anti-pituitary; lines 240 and 243, what means isolated?; line 241, maybe predisposition?; ;line 253, has published; line 261;, has selimitations; add pages in ref 12

Author Response

Reviewer 3

In their manuscript, Levy at al present 17 cases of ICI-induced hypophysitis and suggest means for diagnosis and follow up of ICI-treated cancer patients. The manuscript is interesting and of significant clinical utility. In general it is well structured, although I would definitely suggest that a native English speaker reviews it. Some comments follow.

Major comments

  1. Paragraph 2 in Discussion is quite confusing. I am not sure that the conclusive statement (lines 213-214) is correct, 28% of cases with MRI +/ve results is not negligible. Pls rephrase the entire paragraph and definitely change last sentence.
  2. Do the authors think (and accordingly suggest) that MRIs are useless, else not recommended, in the clinical diagnosis of hypophysitis? Because this is what I understand in paragraph lines 244-260. If so, you need to strongly support this conclusion, with more clinical data and the literature.
  3. Lines 262-264: PDL1 inhibitors are more frequently used than in 1 out of 17 patients. This depends on the type of cancer and the cohort. Pls rephrase. Similarly, I am not convinced that PD1 and PDL1 induced hypophysitis share the same mechanism. These ICIs target completely different cells, signaling Is not the same. Pls correct.
  4. The patient who recovered ACTH deficiency was administered the same dose and scheme of anti-PD1 treatment? Which ICI was used in this particular patient? This may be of relevance and linked to recovery. If possible, compare your data with those in refs 6 and 31.

Minor comments

  1. Line 45, replace the word including
  2. In Materials and Methods add how you determined the hormones listed or include a ref with details on the methodology used.
  3. 1 is not a flow chart, I would say it is the process you followed to select cases reported here
  4. Line 139, pls specify clinical symptoms
  5. Pls look at and correct: line 26, was thought; line 29, 17 cases is not a large cohort, pls change wording; line 36, inhibitor-induced; line 49, treated with; line 60, is now well described; line 62, inhibitor (only 1 approved); line 78, combination; line 89, is it biopsy?; line 91, within two months; line 152, of severity grade; line 170, following diagnosis; line 195, after diagnosis; line 200, in all cases instead of always; line 228, anti-pituitary; lines 240 and 243, what means isolated?; line 241, maybe predisposition?; ;line 253, has published; line 261;, has selimitations; add pages in ref 12

Response to Reviewer 3:

Dear Reviewer,

Thank you for your pertinent comments and suggestions. As suggested, the manuscript has been proofreading by a native English speaker. Please find bellow our point-by-point responses.

Major comments:

  • We have rephrased this paragraph as follow (lines 237 to 247): “The clinical characteristics are similar to those previously described regarding PD1/PDL1 inhibitor-induced hypophysitis, in particular those recently reported by Faje et al in a cohort of 22 PD1 inhibitor-induced hypophysitis [6]. In contrast to that found herein, a pituitary enlargement on MRI was reported in 28% of cases by Faje et al. [6], considering MRIs performed within a month after diagnosis. As a previous study found that pituitary gland enlargement resolved within a month in approximately half of patients with ipilimumab-associated hypophysitis within a month [21], and we considered MRIs performed up to two months, the latest MRIs could have missed the enlargement phase (three MRIs in the present study were performed after one month). However, it can be concluded that MRI abnormality is less common in PD1/PDL1 inhibitor-induced hypophysitis than in CTLA4 inhibitor-induced hypophysitis.”

  • The authors do not think that pituitary MRI is not recommended in the diagnosis of hypophysitis, because it is mandatory to rule out differential diagnosis, particularly pituitary metastasis in these cancer patients. What we suggest is that the absence of MRI abnormality does not exclude the diagnosis of PD1/PDL1 inhibitor-induced hypophysitis. The manuscript has been modified as follow to make it more understandable (lines 234 to 236): “Additionally, MRI was not contributive to positive diagnosis as none showed pituitary abnormality, although it remains necessary to rule out differential diagnoses and in particular pituitary metastasis.”

  • Thank you for this correction. We have rephrased this paragraph as follow (lines 294 to 299): “Only one patient of the cohort received PDL1 inhibitor, which may seem insignificant. This is linked to the fact that the majority of patients were treated for melanoma, for which only PD1 inhibitors were approved in France at the time of the study. This case shows that hypophysitis can also occur with PDL1 inhibitors, and the clinical presentation is similar to PD1 inhibitor-induced hypophysitis, both in previous reports [18] and herein. but it is consistent with clinical practice as PDL1 inhibitors are used less than PD1 inhibitors. Since PD1 and PDL1 are part of the same signaling pathway, we hypothesized that the hypophysitis they induce share the same mechanism.”

  • The patient who recovered ACTH deficiency was receiving pembrolizumab every three weeks at a dose of 2mg/kg. This is the same treatment regimen than other patients treated for melanoma. We have added this information in the text as follow (lines 206 to 208): “One patient recovered his ACTH function two months after diagnosis. This patient was a 52-year-old man treated for metastatic melanoma with pembrolizumab every three weeks at a dose of 2mg/kg, a treatment regimen similar to other patients treated for melanoma. He presented a grade 1 hypophysitis after four cycles of pembrolizumab.”

Comparison with other cases described in the literature was not possible because the patients of ref 6 and 31 received ipilimumab or combination treatment without precision. We added at line 308 ref 22 in which Faje et al. present the patient that recovered of ACTH deficiency induced by ipilimumab.

Minor comments:

  • Line 45, we rephrased the sentence as follow: “Immune checkpoint inhibitors (ICI) target key points in the immune system’s tolerance to tumors to enhance the anti-tumor response, such as including cytotoxic T lymphocyte antigen 4 (CTLA4), programmed cell death 1 protein (PD1) and its ligand programmed cell death 1 ligand 1 (PDL1).”

  • As requested, we have added biological assay methods (lines 97-104): “Serum hormone measurements were performed as follow : cortisol measurement was performed by chemiluminescence immune assay (I2000; Abbot and Aia 360; Tosoh) or electrogenerated chemiluminescence (Elecsys; Roche Diagnostics); ACTH measurement was performed by chemiluminescence immune assay (Liaison XL; DiaSorin) or electrogenerated chemiluminescence (Cobas e601; Roche Diagnostics); TSH, free T4, free T3, FSH and LH measurements were performed by chemiluminescence immune assay (I2000; Abbot); total testosterone measurement was performed by an in-house radioimmunoassay with extraction and chromatography; prolactine measurement was performed by electrogenerated chemiluminescence (Cobas e601; Roche); IGF1 measurement was performed by chemiluminescence immune assay (ISYS; IDS).”

  • Thank you for this correction, we have changed the title of Figure 1 to “Selection process to include cases”.

  • Line 166, clinical symptoms have been added in the text: “The signs that led to the diagnosis of hypophysitis were clinical symptoms (fatigue, nausea and/or loss of appetite) in most cases (76.5%)”.

  • Line 26: correction has been made. Line 29: we have removed the word “large cohort”. Lines 36-42-60-62-78: corrections have been made. Line 90: “histopathological confirmation” has been changed to “biopsy”. Lines 93-180-197-231-259: corrections have been made. Lines 237-276: isolated means that there is no other pituitary deficiency associated with ATCH deficiency. Lines 274-286-294: corrections have been made. Ref 12: pages have been added.

Round 2

Reviewer 3 Report

I am satisfied with the authors' corrections/amendments in text and answers to my comments. The revised version is improved. My suggestion is that the manuscript may now be accepted for publication.